# Controllable Combustion Synthesis of SiC Nanowhiskers in a Si-C-N System: The Role of the Catalyst

**Min Xia [1,*,†], Hong-Yan Guo [2,3,†] and Muhammad Irfan Hussain [1]**

[1]  Institute of Special Ceramics and Powder Metallurgy, University of Science & Technology Beijing, Beijing 100083, China; h.irfan.uaf@Outlook.com

[2]  College of Civil Engineering, Tongji University, Shanghai 200092, China; hongyanguo124@163.com

[3]  State Key Laboratory for GeoMechanics and Deep Underground Engineering, China University of Mining and Technology, Beijing 100083, China

\*  Correspondence: xmdsg@ustb.edu.cn; Tel.: +86-10-62334951

†  These authors contributed equally to this work.

**Abstract:** Silicon carbide (SiC) nanowhiskers (NWs) constitute an important type of optical and structural materials. Herein, SiC NWs were successfully combustion synthesized (CSed) in a Si-C-N system using tungsten (W) as a catalyst. Scanning electron microscopy, transmission electron microscopy, and X-ray diffraction were used to characterize the SiC NWs. Results of morphological characterization indicated that the W-catalyzed CSed SiC NWs products were fluffy from surface to the core, and they were about several hundred micrometers in length with diameters less than 1 μm. For the comprehensive understanding of the initial growing progress of W-catalyzed CSed SiC NWs, the absorption behavior of C, N, and Si atoms on the crystal planes of W (100), W (110), and W (111) surfaces was investigated by using first-principles calculations. The calculated surface energy ($E_{surf}$) of the studied W surfaces and the absorption energy of C, N, and Si atoms on different sites, indicate that the C atom has a priority to sink to the nanometer catalysts grain of W, and the pre-sunk C atom then reacts with Si atom to form NWs.

**Keywords:** optical materials; combustion synthesis; silicon carbide nanowhiskers; first-principles calculations

## 1. Introduction

Silicon carbide (SiC) has excellent thermal conductivity, chemical inertness, and high value of the Young's modulus, strength, toughness, and optics properties [1–4]. Thus, SiC was generally used in high temperature, high power, and high frequency as well as in harsh environments. The outstanding mechanical properties of SiC nanowhiskers (NWs) make it a promising candidate for the reinforcing phase in ceramic, metal, and polymer matrix composites. SiC NWs also show potential for fruitful applications in field emission displays, nanosensors, nanoscale electro-devices, and optoelectronic devices [1–4]. To date, many methods [5–15] have been applied to prepare SiC whiskers or NWs, including the most common and commercial methods such as chemical vapor deposition from silanes, thermal decomposition of rice hulls, and thermal reduction of silicon oxides, in particular on silica and carbon solid mixtures [16]. Though they are industrial production processes, there are still some shortcomings such as the large amount of energy and time consumption, the small amount of whiskers in the whiskers/particle products, and a certain amount of residual Si and C particles and low-melting Fe-Si alloy or other low-melting alloys in the products; most of all, these low-melting particles are detrimental phases in high-temperature environments. To get high quality,

a complicated post process is needed to get high quality nanowhiskers, since whiskers must be separated from the whiskers/particle and purified of the residual Si and C particles and low melting alloys in the products. These shortcomings cause the high production cost directly or indirectly [1]. Combustion synthesis (CS) is an easy, efficient, and low-cost method to obtain a wide variety of materials [1,2,17]. In comparison with the conventional synthesis methods of SiC whiskers, the CS process possesses significant advantages such as great energy efficiency, high purity of the products, and high production rate.

It is well known that an empirical criterion typically adopted for determining the feasibility of CS is an adiabatic temperature exceeding 1800 K [18]. For combustion synthesized (CSed) pure SiC nanowhiskers (NWs), however, the CS process for the formation of pure SiC nanowhiskers (NWs) needs extra energy, due to a low adiabatic combustion temperature of the Si/C system (ca 54 1600–1700 K) [19,20], In this case, introduction of nitrogen can be a feasible approach, since a strong exothermic reaction, $3Si(s) + 2 N_2(g) \rightarrow Si_3N_4(s)$ (calculated adiabatic temperature can reach above 4000 K [21,22]), could be a smart way to assist the CS of SiC. Thus, it is crucial to study the fabrication of high purity SiC phase in the Si-C-N system.

However, many factors can affect the growth of SiC NWs, therefore, control of the morphology of SiC to be one-dimensional (1D) nanostructure in a Si-C-N system is a key issue. Of particular interest is the catalyst, which plays a critical role in the growth of SiC NWs. However, the exact role of the catalyst in whisker growth, depending on the type of selected catalyst, is still unknown and it should be further clarified. Furthermore, not only the effect of the catalyst itself, but also the final status of the catalyst in NWs, in particular in their high temperature applications, should also be taken into account. We have previously reported the fabrication of CSed SiC NWs using titanium powder as a catalyst, but the yield of whiskers needs further optimization [1]. Herein, tungsten (W) was selected as the catalyst because of its high melting point, thus it is not harmful in high temperature structure materials. Moreover, W can react with C, N, or Si elements to form high melting points tungsten carbides, tungsten nitrides, or tungsten silicides, respectively, which is favorable for high temperature performance in structural materials. Based on the experimental data, a comprehensive understanding of the catalyst effect of W-catalyzed CSed one-dimensional SiC nanostructures in a Si-C-N system was achieved through first-principles calculation.

## 2. Experimental

Starting powders were prepared by mixing Si powder (1–2 μm, Beijing Xing Rong Yuan Technology Co., Ltd. Beijing, China), silicon nitride ($Si_3N_4$) powder (prepared in our laboratory with 87% α phase, 1–2 μm grain size, and 99% purity [23]), ammonium fluoride ($NH_4F$, analytical pure, Sinopharm Chemical Reagent Beijing Co., Ltd., Beijing, China), W powder (200 nm, Beijing Xing Rong Yuan Technology Co., Ltd. Beijing, China), and polytetrafluoroethylene (PTFE, Shanghai 3F New Materials Technology Co., Ltd., Shanghai, China) powder (components of the mixtures and corresponding roles in the synthesis process are listed in Table 1). Mixtures were ball milled for 8 h in alcohol using $Si_3N_4$ balls as the medium with the ball/mixture weight ratio of 4:1, then dried and sieved. The sieved mixture was placed in a graphite vessel, then evacuated and backfilled with general industrial nitrogen to ~4–6 MPa. A W heating coil was used to ignite the nitridation reaction system [1,2].

**Table 1.** Components of reactants. PTFE: polytetrafluoroethylene.

| Sample | SC | W-SC | Role |
|---|---|---|---|
| Si (wt.%) | 85 | 85 | Raw materials |
| $Si_3N_4$ (wt.%) | 8 | 8 | Diluent [23] |
| $NH_4F$ (wt.%) | 2 | 2 | Active diluent [23] |
| W (wt.%) | 0 | 5 | Catalyst |
| PTFE (vol.%) | 50 | 50 | Create C element\spatial for whisker growth\supply combustion energy [1] |

The composition and crystal structure of as-synthesized whiskers were investigated by X-ray diffraction (XRD, TTRIII, Rigaku Corporation, Tokyo, Japan). The morphologies were observed by field emission scanning electron microscope (FESEM, SUPRA$^{TM}$ 55, Zeiss, Germany). The intrinsic structure of whiskers was characterized by transmission electron microscopy (TEM, FEI TECNAI G2 F20, FEI, USA).

## 3. Results and Discussions

### 3.1. Characterization of Combustion Synthesized (CSed) Silicon carbide Nanowhiskers (SiC NWs)

Figure 1 shows the XRD patterns of the CSed SiC samples, indicating that the sample without added W particles exhibits the presence of residual C and Si elements. However, when W was introduced as catalyst, no residual C and Si were observed, and all the main peaks agree well with the standard JCPDS 65-0360, which was identified to be cubic $\beta$-SiC (space group F-43m (216), with the lattice constant a = 4.358 Å). The strong intensity and narrow width of the $\beta$-SiC peaks also demonstrate a highly crystalline structure. Other weak peaks in W-catalyzed samples are demonstrated to be $Si_3N_4$ (JCPDS 09-0250) and $Si_2W$ (JCPDS 44-1055) phases. The melting point of the by-product $Si_2W$ was up to 2423 K, according to the Si-W phase diagram [24], the high melting point of $Si_2W$ and $Si_3N_4$ by products is positive for the high temperature application of SiC NWs. XRD results indicate that the W catalyst is able to hinder the formation of C and Si residues. As we know, residual C and Si elements generally result in poor mechanical properties of bulk ceramics at high temperatures [1].

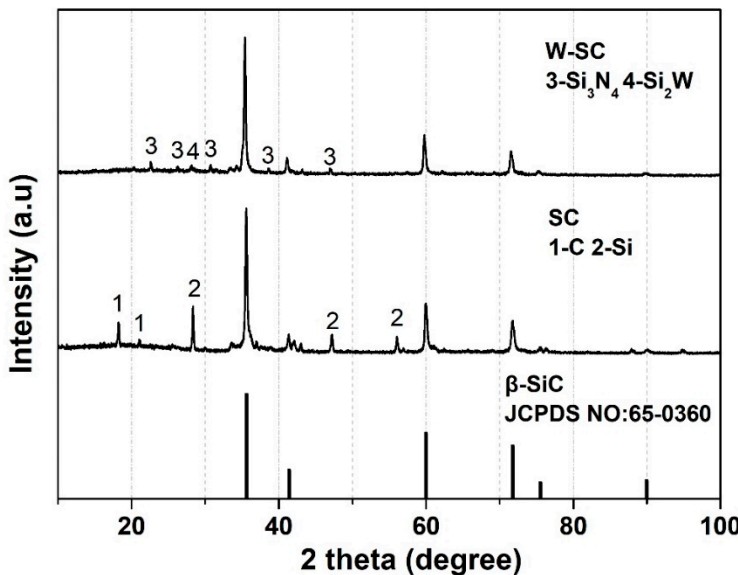

**Figure 1.** X-ray diffraction (XRD) patterns of the combustion synthesized (CSed) silicon carbide nanowhiskers (SiC NWs).

Figure 2a shows the CSed SC sample prepared without W catalyst. Clearly, the sample exhibits a graded structure, that is the core is fluffier than the surface. Moreover, the surface products are somewhat dense (compared with the fluffy core). Figure 2b shows the typical SEM morphology of the SC sample, exhibiting the presence of numerous coarse or spherical residual particles in the SiC NWs, and according to the XRD results shown in Figure 1, these particles are considered to be C and Si particles. In contrast, the W-catalyzed CSed W-SC sample was fluffy from surface to the core, and a little bit green colored, which is similar to another report [25], as shown in Figure 3a. The diameter of the entire product was about 7 cm, which revealed the possibility of large-scale fabrication of CSed W-SC samples. Figure 3b,c display the typical SEM images of the fluffy W-SC sample products, revealing that all products are NW structured, and the CSed W-SC is about several hundred micrometers long,

with diameters less than 1 µm. Tiny amounts of coarse particles may be attributed to the by-products corresponding to $Si_2W$ and $Si_3N_4$ phases.

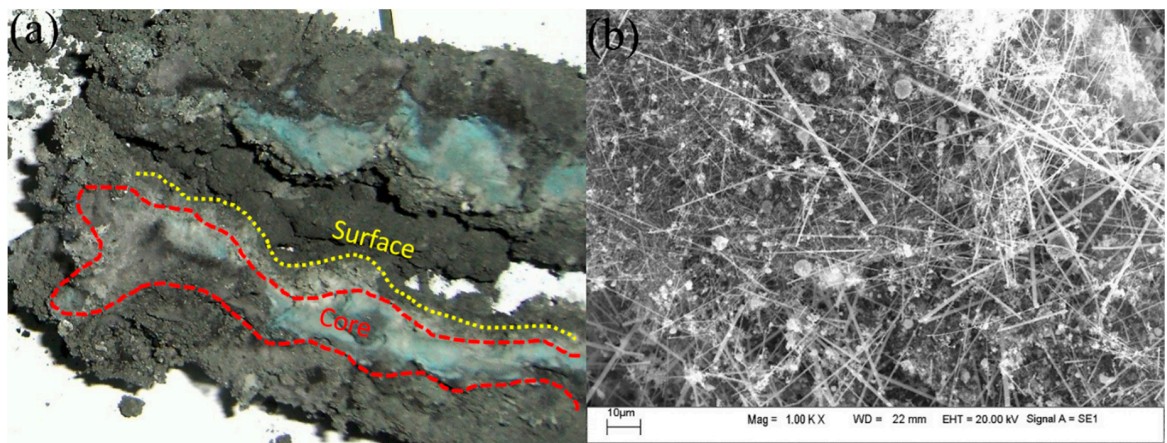

**Figure 2.** CSed SiC NWs without W as the catalyst: (**a**) sample morphology and (**b**) typical scanning electron microscope (SEM) images revealing the existence of residual particles in the NWs.

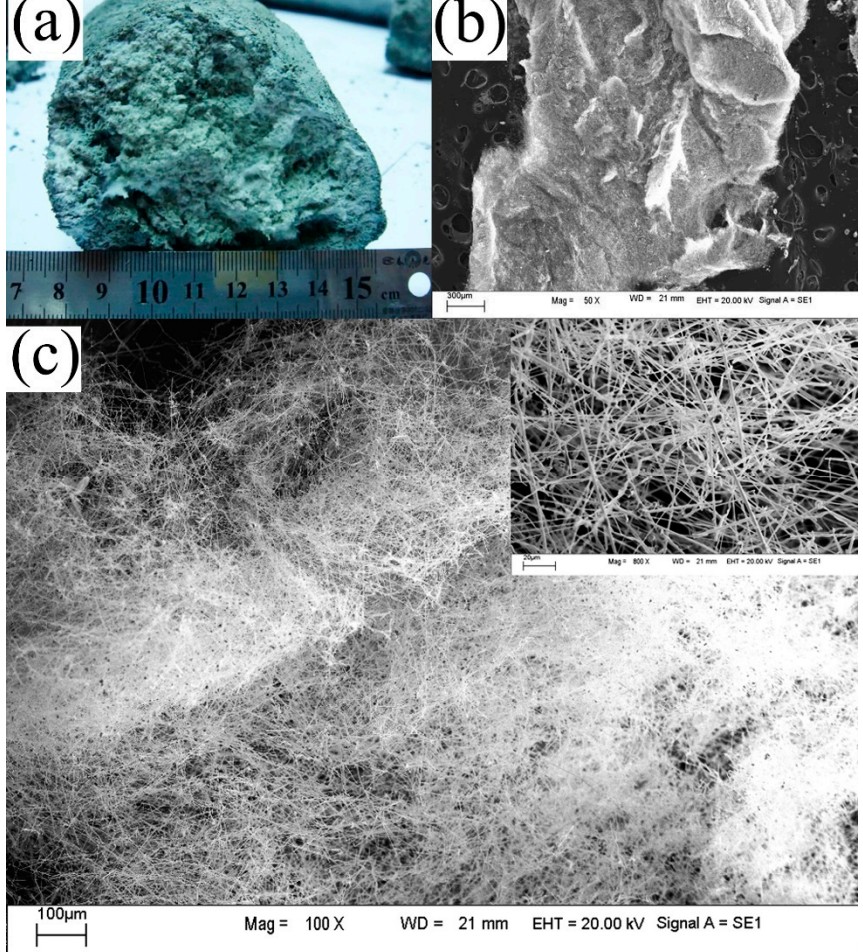

**Figure 3.** (**a**) Fluffy and green colored W-catalyzed CSed SiC NWs; (**b**,**c**) typical SEM images of the W-catalyzed CSed SiC NWs.

Results of TEM provide further insight into the W-catalyzed CSed SiC NWs. Figure 4a shows an individual SiC NW with a diameter of about 20 nm. Moreover, dense stacking faults can be observed

in the whisker, which can be further demonstrated in the atomic structure shown in Figure 4b. The d spacing measured along the growth direction was calculated to be 0.252 nm, indicating the (111) growth direction of the CSed SiC NWs.

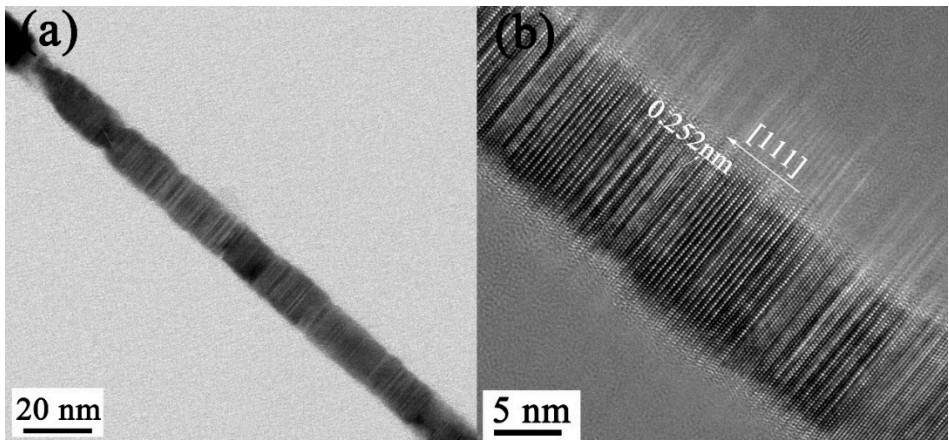

**Figure 4.** (**a**) Transmission electron microscopy (TEM) image of an individual CSed SiC NW and (**b**) Atomic TEM images taken from 4a.

### 3.2. First-Principles Modeling: Adsorption of C, N, and Si Atoms on Different Planes of W Crystal

As mentioned above, W was demonstrated to be an effective catalyst to prepare CSed SiC NWs. However, for better comprehensive understanding, we further explained the formation of SiC NWs in a Si-C-N system, and also the role of N atoms. Moreover, the transformation mode of $Si_3N_4$ and SiC whiskers into pure SiC whiskers with the increase of the amount of C was also explored. It is believed that the absorption sequence and site of the dissociated radical C, N, and Si atoms on the surfaces of the catalyst W particles determines the structure and growth orientation of the NWs. In order to understand the initial growing progress, we investigated the absorption behavior of C, N, and Si atoms on the crystal planes of W (100), W (110), and W (111) surfaces by using the first-principles calculations. The W surfaces were modeled by a seven-layer thick W slab with the middle layer being fixed. The vacuum region between adjacent slabs was 15 Å, which was thick enough to cut off the influence of the neighboring surface. The absorption of C, N, and Si atoms was then investigated by placing a single type of atom on both sides of the W slabs. Figure 5 shows the top views of the studied W surfaces and the possible absorption positions for the dissociated inorganic atoms on the three different W surfaces.

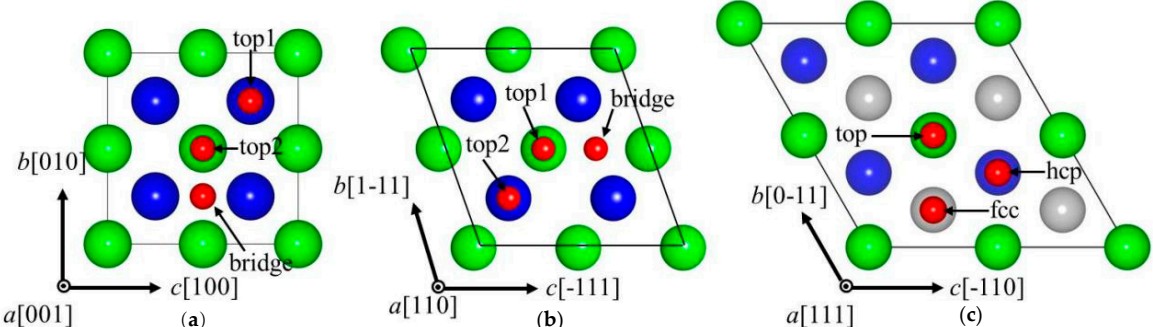

**Figure 5.** Top view of (**a**) 2 × 2 W (100), (**b**) 2 × 2 W (110), and (**c**) 2 × 2 W (111) surfaces (all seven layers). Large green, blue, and gray spheres are for the surface, subsurface, and third layer W atoms, respectively. Small red spheres correspond to the X (X = C, N) atom absorbed at three different sites.

First-principles calculations within the density-functional theory (DFT) framework were carried out by using the Vienna ab initio simulation package (VASP) code [26–29]. The projector augmented

wave (PAW) method [30] was used to describe the electron–ion interaction and the exchange correlation potential between electrons was simulated in the Perdew–Burke–Ernzerhof (PBE) generalized gradient approximation (GGA) form [31]. An energy cutoff of 450 eV was set up for the plane wave basis set and a $(4 \times 4 \times 1)$ $k$-mesh was adopted within the Monkhorst–Pack scheme [32] for sampling the Brillouin zone. All the calculations were carried out with fixed supercell volume and the relaxations were continued until the forces on all atoms converged to less than $10^{-3}$ eV Å$^{-1}$. In this study, the reproduced lattice constant for bcc W by GGA-PBE calculation was 3.17 Å, which is in good agreement with the experimental value [33] and other theoretical results [34]. The calculated surface energy ($E_{surf}$) of the studied W surfaces and the absorption energy of C, N, and Si atoms on different sites are listed in Table 2. The $E_{surf}$ is calculated as follows:

$$Esurf = \frac{1|Eslab - N \cdot Ebulk|}{2Ahkl} \tag{1}$$

where $E_{slab}$ is the total energy of the seven-layer thick slab, $E_{bulk}$ is the energy per W atom in bulk bcc W crystal, $A_{hkl}$ is the area of sectional plane, and $N$ is the number of atoms in the slab model. The adsorption energy ($E_{ad}$) is evaluated by using the following formula:

$$E_{ad} = E_{X/slab} - E_X - E_{slab} \ (X = C, N, \text{and Si}) \tag{2}$$

where $E_X$ and $E_{X/slab}$ are energies of an isolated X ($X$ = C, N, and Si) atom and the slab with X atoms absorbed on its surfaces, respectively. Table 2 shows that among the three absorption atoms, the C atoms show the strongest absorption energy on the W (100) surface ($-11.12$ eV). This indicates that the C atom has a priority to sink to the nanometer catalysts grain of W, and the pre-sunk C atom could react with a Si atom to form NWs. In the case of the W (110) surface, the N atom exhibits the strongest attraction ($-10.03$ eV), which demonstrates the possibility of pre-formation of $Si_3N_4$ NWs on W in a Si-C-N system. Furthermore, the pre-formed $Si_3N_4$ NWs could thus act as an intermediate template to direct the growth of SiC NWs, which was demonstrated by our previous experimental observations [2].

**Table 2.** The calculated surface energy ($E_{surf}$) of the studied W surfaces, and the absorption energy of C, N, and Si atoms on different sites. Different adsorption sites are depicted in Figure 5.

| Surface | $E_{surf}$/(J·m$^{-2}$) | Adsorption Site | $E_{ad}$/eV | | |
|---|---|---|---|---|---|
| | | | C | N | Si |
| W (100) | 3.96 | top1 | $-6.66$ | $-8.63$ | $-4.20$ |
| | | top2 | $-11.12$ | $-10.82$ | $-7.38$ |
| | | bridge | $-8.87$ | $-9.98$ | $-8.15$ |
| W (110) | 3.28 | top1 | $-6.06$ | $-7.55$ | $-4.12$ |
| | | top2 | $-9.31$ | $-10.03$ | $-5.76$ |
| | | bridge | $-8.30$ | $-8.25$ | -4.97 |
| W (111) | 3.76 | top | $-5.21$ | $-7.12$ | $-4.47$ |
| | | hcp | $-6.71$ | $-7.38$ | $-4.73$ |
| | | fcc | $-7.79$ | $-7.36$ | $-6.37$ |

## 4. Conclusions

SiC NWs were successfully obtained via CS in a Si-C-N system using W as a catalyst. The W-catalyzed CSed SiC NWs products were found to be fluffy from surface to the core, and they were about several hundred micrometers long, with diameters less than 1 μm. W could improve the yield of whiskers, and it could decrease the harmful residual Si and C elements. In order to understand the initial growing progress, we investigated the absorption behavior of C, N, and Si atoms on the crystal planes of W (100), W (110), and W (111) surfaces by using first-principles calculations. The calculated $E_{surf}$ of the studied W surfaces and the absorption energy of C, N, and Si atoms on

different sites indicated that the C atom has a priority to sink to the nanometer catalysts grain of W, and the pre-sunk C atom could react with a Si atom to form NWs.

**Author Contributions:** Data curation, H.-Y.G.; Project administration, M.X.; Writing—original draft, M.I.H. All authors have read and agreed to the published version of the manuscript.

**Funding:** This work was supported by the Fundamental Research Funds for the Central Universities (FRF-TP-16-001A3).

**Conflicts of Interest:** The authors declare no conflict of interest.

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
