# Peer review of "Controllable Combustion Synthesis of SiC Nanowhiskers in a Si-C-N System: The Role of the Catalyst"

_applsci, doi:10.3390/app10010252_

Round 1
Reviewer 1 Report
This manuscript deals with the combustion synthesis of SiC nanowhiskers from Si powder (as Si source) and PTFE (as C source) in the presence of Nitrogen gas (necessary for Si powder nitridation), W powder (as a catalyst) and other components.The manuscript is much clearer than in the first submission, and it can be published in the Applied Science journal after a major revision. Here below are my general and detailed comments:
- Some of the authors had already used combustion synthesis in other to prepare SiC nanowhiskers (ref. 1). In that paper they used Ti powder as a catalyst instead of W powder, and all the conditions seem to be identical. However, in this manuscript, no mention was done of the different effect of Ti and W catalysts on the SiC nanowhiskers features (both have the same effect to reduce the Si and C residues, but which is the best catalyst? Is the mechanism the same?). Moreover, although the used preparation methodology was the same, it was named differently. I suggest the authors to clarify this point and discuss any difference/similarity between the rsults comping from the two papers. If the experimental set-up used for the synthesis is identical, this should be specified. Otherwise, the set-up should be reported in some figure.
- The authors used as well Si3N4 and ammonium fluoride in the system, but their role was not specified. Is Si3N4 maybe used due to its templating properties as discussed in a previous work of one of the authors (ref 2)? If yes, please clarify this aspect in the text and discuss adequately, citing ref. 2. In ref.2 nanowires are prepared instead of nanowhiskers. Please evidence any difference and similarities between these two morphologies, chose the same name for the same morphology or clarify this point in the text.
- p2, l74. Please write the exact amount of powder weighted in the combustion mixture and not a range of percentages. For example, is Si 80 or 90 wt%? And is W 1 or 10 wt%? If this is the effective uncertainty of the weighted amount, please specify it somewhere in the text and in the table 1. Moreover, please explain why you decided to use these amount of reagents.
Table 1 Components of reactants Sample |
SC |
W-SC |
Si (wt.%) |
80-90 |
80-90 |
Si3N4 (wt.%) |
1-10 |
1-10 |
NH4F (wt.%) |
1-10 |
1-10 |
W (wt.%) |
0 |
1-10 |
PTFE (vol.%) |
50 |
50 |
- p3, l84-85. I guess that it should be changed into “the high melting point of Si2W and Si3N4 byproducts is positive for the high temperature application of SiC NWs.”
- p3, l86. Why C and Si should be harmful? Please specify it in the text. In the previous paper (ref.1), this was clearly specified in the introduction section (“Though they are industrial production processes, there are still some shortcomings such as large amounts of energy and time consumption, little amount of whiskers in the whiskers/particle products, and a certain amount of residual Si and C particles and low-melting Fe–Si alloy or other low-melting alloys in the products; most of all, these low-melting particles are detrimental phases in high-temperature environment. To get high quality, a complicated post process such as separating whiskers from whiskers/particle products and purifying the whiskers to wash out the residual Si and C particles and low-melting alloys in the products is needed. These shortcomings cause the high production cost directly or indirectly.”): I suggest the authors to write something shorter, citing the previous paper (ref 1). Moreover, I would change the sentence into “XRD results indicate that the W catalyst is able to hinder the formation of C and Si residues.”
- p4, l94. “..the core is fluffier than the surface.”: looking at the cited figure, I can’t say if the core is fluffier than the surface or not. The same is true for the sentence “Moreover, the surface products are 96 somewhat compact. “. Do you have any other evidence? Can you show more details?
- p4, l105. I would change into “Tiny amounts of coarse particles may be attributed to the by-products corresponding to Si2W and Si3N4 phases.”
- p7. L167. Please correct it into “pre-sunk”
Conclusion section is indeed a comment of the results of this specific research. Please try to add one or two sentences on the consequences of your results for example in the synthesis of SiC nanowhiskers and the impact that this results might have in other scientific areas.
- Though some new references were added, the discussion of the obtained results in comparison with the literature is still lacking. Please integrate and enforce the discussion of the obtained results through comparison with other synthesis methodologies, other precursors, other experimental conditions, other catalysts, highlighting similar or different results, in order to place this research in the right scientific context. Why nanowhiskers are a more convenient morphology than other SiC morphologies? Please specify it in the introduction section. This reference might be of help for a comparison, but it is not the only one:
M. Kahar et al; Informacije Midem, Vol. 47, No. 2(2017), 101 – 111
Please try to look for other recent references about SiC that might be important to cite.
Please pay attention to the reference format (author’s names, pages etc)
Author Response
Manuscript ID: applsci-661290
Title: Controllable combustion synthesis of SiC nanowhiskers in Si-C-N system: the role of catalyst
Dear editor
Many thanks for the useful suggestions and comments from reviewers, and many thanks for giving us opportunity to major revise our manuscript, according to the reviewer’s comments, we have revised the manuscript point by point, and changes are red-marked in manuscript. We are looking forward to having your positive response.
Reviewer 2#
This manuscript deals with the combustion synthesis of SiC nanowhiskers from Si powder (as Si source) and PTFE (as C source) in the presence of Nitrogen gas (necessary for Si powder nitridation), W powder (as a catalyst) and other components.The manuscript is much clearer than in the first submission, and it can be published in the Applied Science journal after a major revision. Here below are my general and detailed comments:
- Some of the authors had already used combustion synthesis in other to prepare SiC nanowhiskers (ref. 1). In that paper they used Ti powder as a catalyst instead of W powder, and all the conditions seem to be identical. However, in this manuscript, no mention was done of the different effect of Ti and W catalysts on the SiC nanowhiskers features (both have the same effect to reduce the Si and C residues, but which is the best catalyst? Is the mechanism the same?). Moreover, although the used preparation methodology was the same, it was named differently. I suggest the authors to clarify this point and discuss any difference/similarity between the rsults comping from the two papers. If the experimental set-up used for the synthesis is identical, this should be specified. Otherwise, the set-up should be reported in some figure.
Many thanks for the suggestions, In this Si-C-N combustion system, the high combustion temperature favors high yield of whiskers; in Ti-assisted experiment, we can see clearly that the whiskers is not uniform from the vessel bottom to the surface, the SEM image also indicate it. Even though, the residual Si and C was reduced, but the yield was relatively low (the growth mechanism in Ti-assisted system was mainly considered to be a well-known VLS growth mechanism). In this case, we tried to enhance the combustion temperature to accelerate the growth of whiskers at the bottom of the graphite vessel. The high temperature could also generate radical gaseous Si and C element for Si powders and PTFE powders, and the large amounts of harmful residual Si and C elements could also generated when no W was used. Because the high melting point, W could act as sink site to enhance the growth of whiskers. Thus, the W-assisted whiskers may growth via vapor-solid (VS) growth mechanism.
In all, compared with Ti, W could further improve the yield of whiskers, and could decrease the residual Si and C elements.
The experimental set-up used for the synthesis was specified
- The authors used as well Si3N4 and ammonium fluoride in the system, but their role was not specified. Is Si3N4 maybe used due to its templating properties as discussed in a previous work of one of the authors (ref 2)? If yes, please clarify this aspect in the text and discuss adequately, citing ref. 2. In ref.2 nanowires are prepared instead of nanowhiskers. Please evidence any difference and similarities between these two morphologies, chose the same name for the same morphology or clarify this point in the text.
Many thanks for the suggestions, The Si3N4 “diluent” and ammonium fluoride “active diluent” were all contributed to decrease the combustion temperature in local region between Si and Si powders. If no diluent was used, the in-situ preformed Si3N4 could hinder the diffusion of N into the nearby Si powders, and thus the combustion system may not sustain and large amounts of Si powders were generated. We have add reference in our group.
We have describe the role of raw materials in TABLE 1.
- p2, l74. Please write the exact amount of powder weighted in the combustion mixture and not a range of percentages. For example, is Si 80 or 90 wt%? And is W 1 or 10 wt%? If this is the effective uncertainty of the weighted amount, please specify it somewhere in the text and in the table 1. Moreover, please explain why you decided to use these amount of reagents.
Table 1 Components of reactants Sample |
SC |
W-SC |
Si (wt.%) |
80-90 |
80-90 |
Si3N4 (wt.%) |
1-10 |
1-10 |
NH4F (wt.%) |
1-10 |
1-10 |
W (wt.%) |
0 |
1-10 |
PTFE (vol.%) |
50 |
50 |
Many thanks for the suggestions, We have revised the table. And specified the amounts.
- p3, l84-85. I guess that it should be changed into “the high melting point of Si2W and Si3N4 byproducts is positive for the high temperature application of SiC NWs.”
Many thanks for the suggestions, We have revised it.
- p3, l86. Why C and Si should be harmful? Please specify it in the text. In the previous paper (ref.1), this was clearly specified in the introduction section (“Though they are industrial production processes, there are still some shortcomings such as large amounts of energy and time consumption, little amount of whiskers in the whiskers/particle products, and a certain amount of residual Si and C particles and low-melting Fe–Si alloy or other low-melting alloys in the products; most of all, these low-melting particles are detrimental phases in high-temperature environment. To get high quality, a complicated post process such as separating whiskers from whiskers/particle products and purifying the whiskers to wash out the residual Si and C particles and low-melting alloys in the products is needed. These shortcomings cause the high production cost directly or indirectly.”): I suggest the authors to write something shorter, citing the previous paper (ref 1). Moreover, I would change the sentence into “XRD results indicate that the W catalyst is able to hinder the formation of C and Si residues.”
Many thanks for the suggestions, We have revised the sentence into “XRD results indicate that the W catalyst is able to hinder the formation of C and Si residues. As we know, residual C and Si elements generally resulting in poor mechanical properties of bulk ceramics at high temperatures [1]”.
- p4, l94. “..the core is fluffier than the surface.”: looking at the cited figure, I can’t say if the core is fluffier than the surface or not. The same is true for the sentence “Moreover, the surface products are 96 somewhat compact. “. Do you have any other evidence? Can you show more details?
Many thanks for the suggestions, We have marked figure 2(a) as follows, the surface was indeed compact, which means the low yield of whiskers.
- p4, l105. I would change into “Tiny amounts of coarse particles may be attributed to the by-products corresponding to Si2W and Si3N4 phases.”
Many thanks for the suggestions, We have revised it in manuscript.
- p7. L167. Please correct it into “pre-sunk”
Many thanks for the suggestions, We have revised it in manuscript.
Conclusion section is indeed a comment of the results of this specific research. Please try to add one or two sentences on the consequences of your results for example in the synthesis of SiC nanowhiskers and the impact that this results might have in other scientific areas.
Many thanks for the suggestions, We have added sentences.
- Though some new references were added, the discussion of the obtained results in comparison with the literature is still lacking. Please integrate and enforce the discussion of the obtained results through comparison with other synthesis methodologies, other precursors, other experimental conditions, other catalysts, highlighting similar or different results, in order to place this research in the right scientific context. Why nanowhiskers are a more convenient morphology than other SiC morphologies? Please specify it in the introduction section. This reference might be of help for a comparison, but it is not the only one:
Kahar et al; Informacije Midem, Vol. 47, No. 2(2017), 101 – 111Please try to look for other recent references about SiC that might be important to cite.
Please pay attention to the reference format (author’s names, pages etc)
Many thanks for the suggestions, We have revised the manuscript. And some discussions were also added. And a recent review paper for SiC nanostructures was also referenced. The author’s name was also revised.
Chen, S. and W. Li, et al. (2019). "One-dimensional SiC nanostructures: Designed growth, properties, and applications." Progress in Materials Science 104: 138-214.

Reviewer 2 Report
This manuscript is about the synthesis of silicon carbide nanowhiskers. In the paper the authors characterize the nanowhiskers and look at using tungsten as a catalyst.
Overall, the English language is better than it was in the previous version, however, there are still instances throughout the manuscript where the English language still needs to be improved.
The sentence in lines 43-46 should be improved. Right now it is a bit clunky and hard to understand.
In line 64 the authors mention that they home-made the alpha phase, but there is no reference or explanation as to the synthesis. This should be added.
In line 68 the authors say "nitrogen to a certain pressure." Can the authors be more specific as to what pressure, or what pressure range? If it was ambient pressure generated, then that should be stated.
The authors cite standards in lines 80-83 which are given with a code. Are there ways to reference these? If there are, it would be good to add the references to these lines.
The authors should add estimated standard deviations to the lattice constant in line 81.
In line 86, the authors state that tungsten could significantly decrease the amounts of harmful residual C and Si elements. The authors should provide an explanation for this.
In line 96, the authors should describe what they mean by "grade structure."
In like 164 the authors state that "they possessed uniform length of about several hundred micometers." Can the be more specific about the length, since it is stated the lengths were uniform?
The references in the paper need to be addressed. References 25-28 are listed as the references for VASP, but if you look at the reference list, reference 29 is also a VASP reference.
Author Response
Manuscript ID: applsci-661290
Title: Controllable combustion synthesis of SiC nanowhiskers in Si-C-N system: the role of catalyst
Dear editor
Many thanks for the useful suggestions and comments from reviewers, and many thanks for giving us opportunity to major revise our manuscript, according to the reviewer’s comments, we have revised the manuscript point by point, and changes are red-marked in manuscript. We are looking forward to having your positive response.
Reviewer 1#
This manuscript is about the synthesis of silicon carbide nanowhiskers. In the paper the authors characterize the nanowhiskers and look at using tungsten as a catalyst.
Overall, the English language is better than it was in the previous version, however, there are still instances throughout the manuscript where the English language still needs to be improved.
The sentence in lines 43-46 should be improved. Right now it is a bit clunky and hard to understand.
Many thanks for the suggestions, We have revised the sentence
“however, synthesis of β-SiC from elements by CS process cannot be realized under normal combustion conditions due to a low adiabatic combustion temperature of Si/C system (ca 1600–1700 K) (19, 20), which is too low to sustain the CS reaction. Therefore, this reaction needs extra-energy to sustain itself”
Into
“however, due to a low adiabatic combustion temperature of Si/C system (ca 1600–1700 K) (19, 20), the ynthesis of β-SiC cannot sustain its CS process, , it needs extra-energy to sustain CS process.”
In line 64 the authors mention that they home-made the alpha phase, but there is no reference or explanation as to the synthesis. This should be added.
Many thanks for the suggestions, We have added the reference.
In line 68 the authors say "nitrogen to a certain pressure." Can the authors be more specific as to what pressure, or what pressure range? If it was ambient pressure generated, then that should be stated.
Many thanks for the suggestions,It has been revised into 4-6MPa
The authors cite standards in lines 80-83 which are given with a code. Are there ways to reference these? If there are, it would be good to add the references to these lines.
Many thanks for the suggestions, all these standard codes was check from the standard database in XRD analysis software “JADE”, but not from published references.
The authors should add estimated standard deviations to the lattice constant in line 81.
Many thanks for the suggestions, Sorry for misunderstanding, the lattice constant is the standard value of standard cubic β-SiC, we have revised the bracket to avoid misunderstanding.
Original: which was identified to be cubic β-SiC , (space group F-43m (216)),with the lattice constant a = 4.358 Å
Revised: which was identified to be cubic β-SiC , (space group F-43m (216),with the lattice constant a = 4.358 Å)
In line 86, the authors state that tungsten could significantly decrease the amounts of harmful residual C and Si elements. The authors should provide an explanation for this.
Many thanks for the suggestions, We have revised the sentence into “XRD results indicate that W could significantly decrease the amounts of harmful residual C and Si elements. As we know, residual C and Si elements generally resulting in poor mechanical properties of bulk ceramics at high temperatures. “
In line 96, the authors should describe what they mean by "grade structure."
Many thanks for the suggestions, the graded structure means the core is fluffier than the surface. We have revised the sentence, “the sample exhibits a graded structure, and that is the core is fluffier than the surface”,
In like 164 the authors state that "they possessed uniform length of about several hundred micometers." Can the be more specific about the length, since it is stated the lengths were uniform?
Many thanks for the suggestions, We can not get the accurate length of whiskers from SEM, since the magnification is just about 100x, the individual whiskers were overlapped, thus it is difficult to measure the accurate length, to avoid misunderstanding, we delete “uniform”.
The references in the paper need to be addressed. References 25-28 are listed as the references for VASP, but if you look at the reference list, reference 29 is also a VASP reference.
Many thanks for the suggestions, The projector augmented wave (PAW) method (29) is detailed method in VASP code, we check it, there was no problem on reference 29..

Round 2
Reviewer 1 Report
This revised version of the manuscript is certainly improved and now the scope, the strategy and the results are enough clear to be published in Applied Sciences. The authors’ response to my comments is also mostly satisfactory.
I have just two queries:
1) L 109-110 “Moreover, the surface products are somewhat compact.” PLEASE CLARIFY THIS SENTENCE (I still do not understand the meaning of compact in this context. Is it the contrary of fluffy? Please explain it better).
2) Authors did not reply to this comment "In ref.2 nanowires are prepared instead of nanowhiskers. Please evidence any difference and similarities between these two morphologies, chose the same name for the same morphology or clarify this point in the text."
I have also some minor editing corrections, which I list below in the corrected form:
L 17 as a catalyst
L 26 “pre-sunk”
L 32 “optical properties”; “SiC material was…”(or “SiC materials were…”)
L 44-47 “A complicated post process is needed to get high quality nanowhiskers, since whiskers must be separated from whiskers/particle and purified of the residual Si and C particles and low melting alloys in the products.”
L 49 “In comparison with the conventional synthesis methods of SiC whiskers, CS process possesses significant advantages such as great energy efficiency, high purity of the 50 products, and high production rate.”
L 53-56 “The CS process for the formation of pure SiC nanowhiskers (NWs) needs extra-energy, due to a low adiabatic combustion temperature of Si/C system (ca 54 1600–1700 K) (19, 20).”
L 57 “exothermic reaction 3Si(s) + 2 N2(g) → Si3N4(s)”
L 61-63 “Of particular interest is the catalyst, which plays a critical role in the growth of SiC NWs. However, the exact role of catalyst in whiskers growth, depending on the type of selected catalyst, is still unknown and it should be further clarified.”
L 65-67 “We have previously reported the fabrication of CSed SiC NWs using titanium powder as a catalyst, but the yield of whiskers needs further optimization (1).”
L 70 ..”high temperature performance in structural materials..” (did I understood correctly?)
L 70-73 “Based on the experimental data, a comprehensive understanding of the catalyst effect of W-catalyzed CSed one-dimensional SiC nanostructures in Si-C-N system was achieved through first-principles calculation.”
L 76 “(prepared in our laboratory with 87% α phase, 1–2 μm grain size and 99% purity (23))” (did I understood correctly?)
L 77-78 “(PTFE) powder. Components of the mixtures and corresponding roles in the synthesis process are listed in Table 1.”
L 93-94 “The strong intensity and narrow width of the β-SiC peaks also demonstrate a highly crystalline structure.”
Author Response
Manuscript ID: applsci-661290
Title: Controllable combustion synthesis of SiC nanowhiskers in Si-C-N system: the role of catalyst
Dear editor
Many thanks for the useful suggestions and comments from reviewers, and many thanks for giving us opportunity to further revise our manuscript, according to the reviewer’s comments, we have revised the manuscript point by point, and changes are red-marked in manuscript. We are looking forward to having your positive response.
Reviwer 1#
This revised version of the manuscript is certainly improved and now the scope, the strategy and the results are enough clear to be published in Applied Sciences. The authors’ response to my comments is also mostly satisfactory.
I have just two queries:
L 109-110 “Moreover, the surface products are somewhat compact.” PLEASE CLARIFY THIS SENTENCE (I still do not understand the meaning of compact in this context. Is it the contrary of fluffy? Please explain it better).
Sorry for it. In fact, the “compact” here was actually the contrary of fluffy.
We have revised the sentence into “the surface products are somewhat dense (compared with the fluffly core)”
Authors did not reply to this comment "In ref.2 nanowires are prepared instead of nanowhiskers. Please evidence any difference and similarities between these two morphologies, chose the same name for the same morphology or clarify this point in the text."
Sorry for it, and many thanks. To the best of our knowledge, there was no sharp difference between nanowire and nanowhiskers, maybe whiskers are generally straight in length direction. Generally, we describe nanowhiskers to be nanowire when most of the diameter of whiskers are below 100nm (nanowhiskers 100nm to <1000nm; and nanowires <1000nm).
So in ref.1 we also use nanowhiskers. Since the diameters of the observed whiskers are below 100nm in ref.2, we described it as nanowire. In these cases, we describe it as nanowhiskers.
I have also some minor editing corrections, which I list below in the corrected form:
L 17 as a catalyst
Revised
L 26 “pre-sunk”
Revised
L 32 “optical properties”; “SiC material was…”(or “SiC materials were…”)
Revised
L 44-47 “A complicated post process is needed to get high quality nanowhiskers, since whiskers must be separated from whiskers/particle and purified of the residual Si and C particles and low melting alloys in the products.”
Revised
L 49 “In comparison with the conventional synthesis methods of SiC whiskers, CS process possesses significant advantages such as great energy efficiency, high purity of the 50 products, and high production rate.”
Revised
L 53-56 “The CS process for the formation of pure SiC nanowhiskers (NWs) needs extra-energy, due to a low adiabatic combustion temperature of Si/C system (ca 54 1600–1700 K) (19, 20).”
Revised
L 57 “exothermic reaction 3Si(s) + 2 N2(g) → Si3N4(s)”
Revised
L 61-63 “Of particular interest is the catalyst, which plays a critical role in the growth of SiC NWs. However, the exact role of catalyst in whiskers growth, depending on the type of selected catalyst, is still unknown and it should be further clarified.”
Revised
L 65-67 “We have previously reported the fabrication of CSed SiC NWs using titanium powder as a catalyst, but the yield of whiskers needs further optimization (1).”
Revised
L 70 ..”high temperature performance in structural materials..” (did I understood correctly?)
YES, it should be structural. Many thanks for the careful comments
L 70-73 “Based on the experimental data, a comprehensive understanding of the catalyst effect of W-catalyzed CSed one-dimensional SiC nanostructures in Si-C-N system was achieved through first-principles calculation.”
Revised
L 76 “(prepared in our laboratory with 87% α phase, 1–2 μm grain size and 99% purity (23))” (did I understood correctly?)
Yes, it will be better like this. We have revised it in manuscript.
L 77-78 “(PTFE) powder. Components of the mixtures and corresponding roles in the synthesis process are listed in Table 1.”
Revised
L 93-94 “The strong intensity and narrow width of the β-SiC peaks also demonstrate a highly crystalline structure.”
Revised
Many thanks for your professional technical comments and careful suggestions, and many thanks for give us chance to grow with u. Hope to have a face to face chance to learn from u. I will show my highest respect. THANKS a lot.

Reviewer 2 Report
This manuscript is about the synthesis of silicon carbide nanowhiskers. In the paper the authors characterize the nanowhiskers and look at using tungsten as a catalyst.
This is a much improved manuscript from the original submission. There are a few grammatical errors that I have found, which should be addressed, and after that, I feel that the manuscript should be accepted.
Some examples (not all) of grammatical errors to fix:
Line 55 - 'ynthesis' should be corrected to 'synthesis'
Line 55 - ', ,' should be corrected to ','
Line 56- 'porcess' should be corrected to 'process'
Line 69 - 'silicide' should be corrected to 'silicides'
Line 177 - 'length' should be changed to 'lengths'
Author Response
Manuscript ID: applsci-661290
Title: Controllable combustion synthesis of SiC nanowhiskers in Si-C-N system: the role of catalyst
Dear editor
Many thanks for the useful suggestions and comments from reviewers, and many thanks for giving us opportunity to further revise our manuscript, according to the reviewer’s comments, we have revised the manuscript point by point, and changes are red-marked in manuscript. We are looking forward to having your positive response.
Reviwer 2#
This manuscript is about the synthesis of silicon carbide nanowhiskers. In the paper the authors characterize the nanowhiskers and look at using tungsten as a catalyst.
This is a much improved manuscript from the original submission. There are a few grammatical errors that I have found, which should be addressed, and after that, I feel that the manuscript should be accepted.
Some examples (not all) of grammatical errors to fix:
Line 55 - 'ynthesis' should be corrected to 'synthesis'
Line 55 - ', ,' should be corrected to ','
Line 56- 'porcess' should be corrected to 'process'
The whole sentence in line 55-56 have been revised.
Line 69 - 'silicide' should be corrected to 'silicides'
Line 177 - 'length' should be changed to 'lengths'
Many thanks for the careful suggestions, all grammatical errors have been revised.

This manuscript is a resubmission of an earlier submission. The following is a list of the peer review reports and author responses from that submission.
Round 1
Reviewer 1 Report
This manuscript is about the synthesis of SiC nanowhiskers by solid-state combustion synthesis under nitrogen atmosphere. XRD, SEM and TEM were used as characterization techniques for the SiC investigation. First-principles calculations were also performed in order to evaluate the affinity of the Si, C and N atoms for the W surface.
The formation of SiC whiskers is interesting, although it is not new. The use of W powder as a catalyst for SiC production is original, but please notice that tungsten in the form of oxide was already employed as catalyst for SiC production. The authors previously used a Ti catalyst with the same process. The association of the experiment with the first principle calculation is interesting, although theoretical simulation it should be better integrated and discussed within the literature panorama. However, I do not recommend this manuscript for publication in Applied Sciences mainly because it needs to be re-written in order to be readable and the authors should do a deep revision before the next submission to any scientific journal. Some specific reasons of the rejection are reported below:
- English should be revised all over the manuscript. The section dedicated to the first-principles calculations is written in a slightly better English. However, the whole manuscript is very hard to read and understand.
- In addition, experimental part should be integrated with enough details (amounts, temperature, instrumentation used, procedure etc…) for a reader to reproduce the same experiment.
- A more detailed discussion of the synthesis procedure and of the obtained results is required.
- A detailed comparison with the existing literature should be performed and the advantages of this procedure with respect to other ones should be highlighted.
- Reference to the scientific literature is not up-to-date. A bibliographic search should be done on the recent literature papers on the subject.
Here below I report few interesting literature works:
Materials Science-Poland, 34(4), 2016, pp. 770-779 (http://www.materialsscience.pwr.wroc.pl/) DOI: 10.1515/msp-2016-0101 Mater Sci: Mater Electron (2014) 25:5302–5308, DOI 10.1007/s10854-014-2305-4 C H Voon et al 2016 IOP Conf. Ser.: Mater. Sci. Eng. 160 012057 Materials Chemistry and Physics 144 (2014) 560-567 Journal ofCrystalGrowth453(2016)7–12 Progress in Solid State Chemistry 43 (2015) 98-122 Molecules 2017, 22(11), 2033; https://doi.org/10.3390/molecules22112033 Fan J., Chu P.K. (2014) SiC Nanowires. In: Silicon Carbide Nanostructures. Engineering Materials and Processes. Springer, Cham Applied Surface Science 359 (2015) 177–187- The formation of W-containing secondary phases should be commented more in details.
- In order to evidence the role of catalyst a reference material should be prepared without using any W catalyst.
- page 2, line 65: please change “nm” into Å
- page 2, line 64: Please specify the space group of the cubic SiC structure
Reviewer 2 Report
This manuscript describes the synthesis of SiC nanowiskers along with their characterization and some computational studies on the absorption of Si, C, and N atoms at different sites.
Overall, the grammar needs to be majorly improved before the manuscript is published. THere are also some missing subscripts (line 54, line 171).
Additionally, the authors should discuss the PXRD pattern further, and specifically state what other phases are present in the PXRD pattern. Some of the peaks that belong to the standard are very small in the as synthesized SiC NWs. I agree that they are in the pattern, but there are several other peaks (for example, the 4 peaks before 35, and the small speaks around the large peak at 35 and 41) that are at a similar intenisty which do not belong to the standard.
The authors should also comment on why there is a green color to the SiC NWs (line 72). It is my thought that the synthesized NW's would be white, since there aren't any metals involved to give a color other than white. Is the green from some sort of impurity?
The authors should double check the VASP reference (line 175). When referencing VASP, you need to include four different references (at least, if I am not mistaken).
Reviewer 3 Report
The present article is devoted to the synthesis of SiC nanowhiskers using tungsten as catalyser. The authors analyse through DFT calculations the absorption from different faces of the W catalyst of C, N and Si atoms to, by means of combustion, achieve the SiC nanowhiskers. The role of N atoms is to increase the temperature through exothermic reaction with Si atoms.
The article is well presented; however, some changes will improve notably the quality of the contribution.
In the experimental section, the authors must define the quantity of each reactive: Si powders, Si3N4 powders, NH4F, W powders PTFE powders, since the article is devoted to the synthesis procedure. They also should explain what PTFE is for the readers.
In the characterization of Csed SiC NWs, it is also needed a Le Bail fit or Rietvelt refinement to proof the quality of the product. Moreover, a fit can show the agreement of the peaks of the impurity with the SixW phases noted by the authors, and with that deduce the purity of the product.
In Figure 4 the authors must explain better the schemes, like the flags in the axes. In the figure caption (a) and (c) are the same surfaces.
In line 101 “Figure 5” must be “Figure 4”.
There some typos and the English should be corrected. Some examples are:
Line 46 “…catalyst plays an critical…” should be “…catalyst plays a critical…”
Line 48 “…as catalyst to can accelerate…” should be “…catalyst to accelerate…”
Line 92 “what the role of N atoms?” should be “what is the role of N atoms?”
Line 92 “…in Si-C-N system,” should be “… in Si-C-N system?”
…